# HERMES: TEMPORAL-COHERENT LONG-FORM UNDERSTANDING WITH EPISODES AND SEMANTICS

## ABSTRACT

Existing research often treats long-form videos as extended short videos, leading to several limitations: inadequate capture of long-range dependencies, inefficient processing of redundant information, and failure to extract high-level semantic concepts. To address these issues, we propose a novel approach that more accurately reflects human cognition.[1] This paper introduces **HERMES**: temporal-co**HER**ent long-for**M** understanding with **E**pisodes and **S**emantics, a model that simulates episodic memory accumulation to capture action sequences and reinforces them with semantic knowledge dispersed throughout the video. Our work makes two key contributions: First, we develop an Episodic COmpressor (ECO) module that efficiently aggregates crucial representations from micro to semi-macro levels, overcoming the challenge of long-range dependencies. Second, we propose a Semantics reTRiever (SeTR) that enhances these aggregated representations with semantic information by focusing on the broader context, dramatically reducing feature dimensionality while preserving relevant macro-level information. This addresses the issues of redundancy and lack of high-level concept extraction. Extensive experiments demonstrate that HERMES achieves state-of-the-art performance across multiple long-video understanding benchmarks in both zero-shot and fully-supervised settings. Our code will be made public.

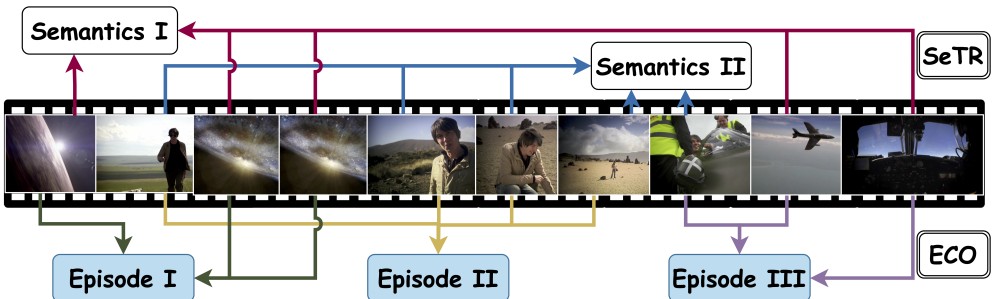

Figure 1: **Semantic Knowledge and Episodic Memory Aggregation:** Our Episodic COmpressor (ECO) processes input video frames and aggregates key episodes: (I) A cosmic setting with a planet and starfield, (II) A narrator explaining the scene, and (III) An aircraft viewed from inside and out. Concurrently, our Semantics reTRiever (SeTR) identifies high-level semantic cues throughout the video, including: (I) The theme of space exploration and (II) Human interaction with technology. This dual-level approach enables comprehensive video understanding by capturing both specific events and overarching concepts.

## 1 INTRODUCTION

Video understanding reflects how humans perceive the world through one of our most essential senses, sight, and drives a wide range of visual and multimodal applications. Whether we want

---

[1]We elaborate on this in Appendix A.7

to create better video summarization tools, index and retrieve specifics from the vast and ever-expanding array of video content, or improve content moderation and copyright enforcement, we need models that excel at video understanding. This requirement extends beyond short videos with few frames—a task that image models can already handle adequately—to encompass the analysis of extended video content spanning minutes and comprising thousands of interrelated frames.

Long-form video understanding is challenging for several reasons. First and foremost is the temporal complexity, as it requires handling a large number of frames throughout the video. Second, it requires a semantic understanding of high-level concepts as well as the narrative structure. The third challenge is the memory and computational constraints, making it non-trivial to solve the previous two challenges. Attempts to address these issues have been made by researchers who mainly borrow ideas from short videos (Wu & Krahenbuhl, 2021; Miech et al., 2020), which is a more mature area of research encompassing action recognition and video classification, among others, and for which datasets are more abundant. However, these approaches, which adopt techniques such as pooling (Faure et al., 2023), or 3D convolutions (Feichtenhofer et al., 2019), often do not fully account for the unique characteristics of long videos that distinguish them from a simple concatenation of short video segments. Some ideas about short-video modeling, especially for those at the spatial level, may also hold for longer ones, but when it comes to long-term modeling, macro-level representations should be extracted efficiently.

In video understanding, we can distinguish between two types of information: episodic and semantic. Episodic information refers to specific, sequential events that occur in a video, while semantic information encompasses overarching themes and concepts. To illustrate, imagine walking through a scene at a birthday party. Episodic information might include observing five people singing a birthday song, followed by one person cutting a cake. These are specific, time-bound events. In contrast, semantic information might involve recognizing decorations scattered throughout the scene, instantly comprehending that you're witnessing a birthday party. This high-level understanding provides a concise overview of the scene and actions, transcending specific moments. Building on these concepts, we propose *temporal-coHERent long-forM understanding with Episodes and Semantics (HERMES)*, a model designed to capture both episodic and semantic information in long-form videos. HERMES comprises two key components: ECO and SeTR. The **E**pisodic **CO**mpressor (**ECO**) aggregates key contiguous information as we process the video, shaping the model's understanding of the scene sequentially without cluttering it. Complementing this, the **SE**mantic re**TR**iever (**SeTR**) identifies and extracts high-level cues that provide a concise overview of the scene and actions. HERMES achieves state-of-the-art performance on four long-form video understanding benchmarks in both zero-shot and fully-supervised settings, notably outperforming the state-of-the-art by a significant 7.3% on LVU(Wu & Krahenbuhl, 2021) and 14.9% on MovieChat-1k (Song et al., 2024).

Our key contributions are as follows:

- We address the challenge of efficiently processing and understanding long-form videos, a problem that has been hindered by computational constraints and the difficulty of capturing complex temporal dynamics over extended durations.

- We propose an Episodic COmpressor (ECO) to stream through the video and keep important episodes by aggregating similar scenes. ECO enables efficient processing of extended video sequences while preserving temporal coherence and narrative structure.

- We develop a Semantics reTRiever (SeTR) that enhances the model's understanding of long videos by distilling high-level semantic cues, providing a cohesive framework for understanding the context and themes within long-form videos.

Through comprehensive evaluation across multiple benchmarks and detailed ablation studies, we validate the effectiveness of ECO and SeTR, and demonstrate their complementary roles in enhancing long-form video understanding

## 2 PROBLEM SETTING

Given a long video $V = \{f_1, f_2, \ldots, f_N\}$, where $f_i$ represents the $i$-th frame and $N$ is the total number of frames, our objective is to develop a model $\mathbf{M}$ that can efficiently process $V$ and construct

an internal understanding $U$ of its content. This understanding should enable the model to answer queries $Q$ or follow instructions $I$ related to the video content. Formally, we aim to optimize the function:

$$\mathbf{M} : (V, I) \to U \tag{1}$$

such that:

- $U$ captures both episodic and semantic information from $V$.
- $U$ can be used to maximize the probability $P(A|Q, U)$ of generating correct answers $A$ to queries $Q$ about the video content.

The key challenges in this formulation are:

- **Temporal Complexity:** Efficiently processing $N$ frames, where $N$ can be very large (thousands of frames for minutes-long videos).
- **Semantic Understanding:** Extracting high-level concepts and narrative structure from the video content.
- **Memory Constraints:** Developing a method that can maintain relevant information without exhausting computational resources.

We aim to address these challenges, by proposing two key contributions:

1. **Episodic COmpressor (ECO):**

$$ECO : f_1, f_2, \ldots, f_N \to e_1, e_2, \ldots, e_K \tag{2}$$

   where $K \ll N$, and $e_i$ represents a compressed episodic memory.

2. **Semantics reTRiever (SeTR):**

$$SeTR : \{f_1, f_2, \ldots, f_N\} \to \{s_1, s_2, \ldots, s_L\} \tag{3}$$

   where $L \ll N$, and $s_i$ represents extracted semantic knowledge.

The final understanding $U$ is then generated by combining the outputs of ECO and SeTR:

$$U = G(ECO(V, I), SeTR(V)) \tag{4}$$

where $G$ is a function that integrates episodic and semantic information. This formulation allows us to approach long-form video understanding in a way that more closely mimics human cognition, addressing the identified challenges while maintaining computational efficiency.

## 3 PROPOSED FRAMEWORK: HERMES

This paper is not about a new LLM or a new way to fine-tune existing LLMs or VLMs. It focuses on leveraging what we know about how humans understand visual scenes to guide the model through the same process. Although this work uses an LLM for autoregressive prediction, the core ideas of episodic memory compression (ECO) and semantic knowledge retrieval (SeTR) can be applied to other models where learning contiguous sequences and high-level representations is advantageous.

Given a video, short or long, and a set of instructions specifying what to do with the video, our method can return the specified output, such as video question answering (VQA) or video classification. It achieves this by leveraging two important properties of human understanding of scenes: episodic memory, which involves determining and stratifying a sequence of frames with similar properties, and semantic knowledge, which can help answer broad questions about the scene (e.g., does it occur at night or during the day?). We refer to the former as ECO, detailed in Section 3.2, and to the latter as SeTR, described in Section 3.4.

### 3.1 WINDOW ENCODER

Our model takes as input a video of arbitrary length. To batch process the video, we first specify a number of frames $N$ to extract, leading to $\mathbf{v} = \{\mathbf{f}_1, \mathbf{f}_2, \ldots, \mathbf{f}_N\}$, where $\mathbf{f}_t$ denotes the $t$-th frame. The

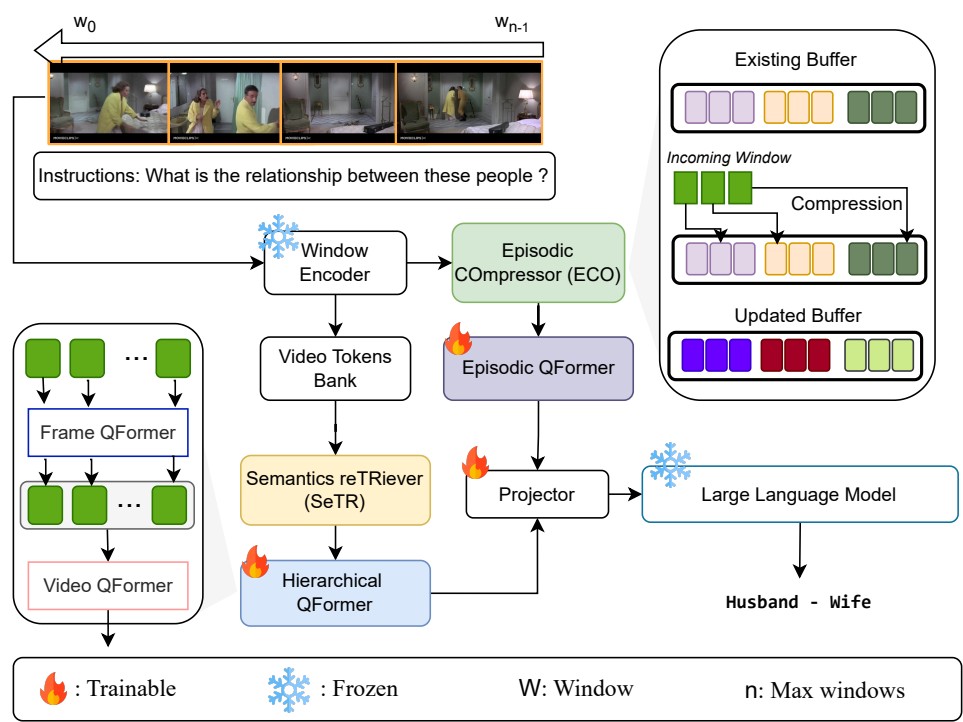

Figure 2: **HERMES framework overview:** We stream through a video window-by-window and extract features using a frozen ViT. Each window feature is processed by the Episodic COmpressor (ECO) in an online fashion, discarding redundancies along the way and retaining video episodes that are passed to an episodic Q-Former. The video token bank contains the concatenated features of every window, and SeTR selects only the high-level information to pass to a hierarchical frame-to-sequence Q-Former. The episodic and high-level representations are then concatenated before being fed to the frozen LLM, which outputs a text following the instructions.

ViT-G/14 encoder (Fang et al., 2023) progressively encodes non-overlapping windows of the video data. The window size $w$ is a divisor of $N$ and determines how many frames to encode at once. The features of each window are denoted as $\mathbf{F}_{w,i} \in \mathbb{R}^{B \times w \times T \times C}$, where $\mathbf{F}_{w,i}$ are the extracted features for the $i$-th window, $B$ the batch size, $T$ the number of visual tokens, and $C$ the number of channels. $\mathbf{F}_w$ are then passed on to the Episodic COmpressor (ECO) described in Section 3.2.

## 3.2 ECO: EPISODIC COMPRESSOR

The proposed Episodic COmpressor (ECO) aggregates the video frames into episodes. This module maintains a memory buffer with a maximum number of episodes $E$. Upon receiving a window of frame features, $F_w$, we first check whether the buffer $\mathcal{M}$ has sufficient bandwidth to support the incoming features. If it does, we simply concatenate them to the buffer; otherwise, we proceed with the compression. At its core, ECO is a distribution process that determines the episode to which a certain frame belongs. It can be summarized as:

---

**Algorithm 1** ECO: Episodic COmpressor

1: $\mathcal{A} \leftarrow \mathcal{M} \oplus F_w$
2: **while** $\|\mathcal{A}\| > E$ **do**
3: $\quad (i^*, j^*) \leftarrow \arg\max_{i \neq j} \frac{\mathcal{A}_i \cdot \mathcal{A}_j}{\|\mathcal{A}_i\| \|\mathcal{A}_j\|}$
4: $\quad \mathcal{A}_{i^*} \leftarrow \frac{(\mathcal{A}_{i^*} + \mathcal{A}_{j^*})}{2}$
5: $\quad \mathcal{A} \leftarrow \mathcal{A} \setminus \mathcal{A}_{j^*}$
6: **end while**
7: $\mathcal{M} \leftarrow \mathcal{A}$

---

$$\mathcal{M} = \begin{cases} \mathcal{M} \oplus F_w & \text{if } \|\mathcal{M}\| + \|F_w\| \leq E \\ \text{ECO}(\mathcal{M}, F_w) & \text{otherwise} \end{cases} \tag{5}$$

Where $\oplus$ is the concatenation operation, $\|\mathcal{M}\|$ and $\|F_w\|$ the sizes of the buffer and the incoming features, respectively.

ECO works as Algorithm 1. As long as concatenating the new window and the buffer results in a size greater than $E$, we compute the cosine similarity between each pair of frame features in $\mathcal{M} \oplus F_w$. We then iteratively merge the most similar frames until the size constraint $E$ is satisfied. Specifically, where $\mathcal{M}$ is the existing buffer, $F_w$ represents the incoming window of frame features, $\mathcal{A}$ is the concatenated buffer and new window, and $\|\mathcal{A}\|$ is the size of $\mathcal{A}$. To summarize Algorithm 1, $\frac{\mathcal{A}_i \cdot \mathcal{A}_j}{\|\mathcal{A}_i\|\|\mathcal{A}_j\|}$ computes the cosine similarity between frame features $\mathcal{A}_i$ and $\mathcal{A}_j$, $\arg\max_{i \neq j}$ finds the pair of frames with the highest cosine similarity, $\frac{(\mathcal{A}_{i*} + \mathcal{A}_{j*})}{2}$ combines the most similar frames, and $\mathcal{A} \setminus \mathcal{A}_{j*}$ removes the frame $\mathcal{A}_{j*}$ from $\mathcal{A}$ after merging. The process repeats until the size of $\mathcal{A}$ is within the maximum allowed episodes $E$.

Similarities can be drawn with He et al. (2024), where cosine similarity serves as the basis for frame reduction. However, their approach is notably inefficient and less intuitive. For a buffer of size $S$, they iterate $S$ times until the buffer reaches capacity, after which each new incoming frame is compared against every other frame in the buffer.

### 3.3 EPISODIC Q-FORMER

The Episodic Q-Former uses the same architecture as the original Q-Former (Li et al., 2023a) and is loaded with weights pretrained by Dai et al. (2023). However, we insert ECO as a pruning module within the Q-Former to combine and branch queries into episodes over the long video. Given initial queries and instructions, we perform self-attention on these queries and then cross-attention between the queries and the visual representations $\mathcal{M}$. The enhanced queries then undergo an ECO-like process, where we iteratively merge similar queries across video windows, effectively forming video query episodes of high information density. The following equation summarizes the process,

$$Q = \mathrm{ECO_q}\left(\mathrm{CA}\left(\mathrm{SA}(Q_0), \mathcal{M}\right)\right) \tag{6}$$

where $Q_0$ represents the initial queries, $\mathcal{M}$ denotes the visual representations from the visual ECO, $\mathrm{SA}(Q_0)$ applies self-attention on the initial queries, and $\mathrm{CA}(\cdot, \mathcal{M})$ performs cross-attention between the self-attended queries and the visual representations. Finally, $\mathrm{ECO_q}(\cdot)$ – note the q to differentiate it from the visual ECO – applies the iterative merging process similar to the visual compression detailed in Section 3.2 on the enhanced queries. The episodic Q-Former outputs $Q \in \mathbb{R}^{B \times q \times C'}$ with $B$, $q$ and $C'$ alluding to the batch size, the number of queries and the channel dimension, respectively.

### 3.4 SETR: SEMANTICS RETRIEVER

To complement ECO and capture higher-level semantic information from the video, we develop a Semantics reTRiever (SeTR). SeTR is designed to identify and consolidate important high-level information that may be scattered (contiguously or not) throughout the video. Given a video feature tensor $F \in \mathbb{R}^{B \times N \times T \times C}$, where $B$ is the batch size, $N$ the number of frames, $T$ the number of tokens per frame and $C$ the channel dimension, SeTR operates as follows: we first normalize $F$ to ensure consistent scaling across features. Second, we apply a stride of $k$ to create two groups, group $X$ containing every $k$-th frame, resulting in $\frac{N}{k}$ frames and group $Y$ with the remaining $N - \frac{N}{k}$ frames. Third, we calculate dot product similarity scores between frames in $X$ and $Y$. Finally, for each frame in $Y$, we merge it with its most similar frame in $X$, based on the computed scores by taking their mean.

This process effectively reduces the number of frames from $N$ to $\frac{N}{k}$, consolidating semantic information while maintaining the most relevant features across the video timeline. The resulting semantic representations are denoted as $F' \in \mathbb{R}^{B \times \frac{N}{k} \times T \times C}$. We evaluate the effectiveness of this approach in Section 4.3. While ToMe (Bolya et al., 2022) have explored token reduction in vision transformers, their approach and objectives differ significantly from ours. Their method focuses on minor token reductions within individual frames, specifically between different layers of a Vision Transformer. In contrast, SeTR retains the most salient frames while significantly reducing redundancies.

### 3.5 HIERARCHICAL QFORMER

Following our SeTR, is a hierarchical Q-Former composed of a frame Q-Former ($fQFormer$), a frame-to-sequence adapter and a video Q-Former ($vQFormer$). The frame Q-Former enhances each semantic piece of information, independently of the others, and the video Q-Former consolidates them. The resulting query $Q_{sem} \in \mathbb{R}^{B \times q \times C'}$ contains the semantic representations of the entire video.

$$Q_{sem} = vQFormer(Linear(fQFormer(F')))  \tag{7}$$

### 3.6 FROM REPRESENTATIONS TO NATURAL LANGUAGE

After obtaining the episodic representations $Q$ and the semantic representations $Q_{sem}$, we prepare them for input into a Large Language Model (LLM). Specifically, we concatenate $Q$ and $Q_{sem}$ to form a unified representation vector. This concatenated vector is then projected into the input embedding space of the LLM using a learned linear transformation. In our implementation, we utilize a Vicuna-7B model (Chiang et al., 2023) as LLM. The model, conditioned on this projected representation and guided by task-specific instructions, generates the requested natural language output. This approach allows us to leverage the LLM's pretrained knowledge and language generation capabilities while incorporating our task-specific episodic and semantic information. The process is summarized by the following equation:

$$\hat{Y} = \text{LLM}(W[Q; Q_{sem}] + b, I)  \tag{8}$$

where $\hat{Y}$ is the generated output, $[Q; Q_{sem}]$ denotes the concatenation of $Q$ and $Q_{sem}$, $W$ and $b$ are the learned projection matrix and bias respectively, and $I$ represents the task-specific instructions.

## 4 EXPERIMENTS

### 4.1 DATASETS AND EVALUATION METRICS

We evaluate our approach on two primary tasks: long-form video classification and long-form video question answering.

For long-form video classification, we utilize three diverse datasets. The first, *LVU* (Wu & Krahenbuhl, 2021), focuses on movie content, offering a rich source of narrative and thematic video data. The second, *Breakfast* (Tang et al., 2019), consists of instructional videos that emphasize procedural and step-by-step understanding. Lastly, *COIN* (Kuehne et al., 2014) is another instructional video dataset, but it covers a broader range of procedural activities compared to Breakfast. We report top-1 classification accuracy on these datasets.

For long-form video question answering, we employ the *MovieChat-1k* dataset (Song et al., 2024) and report both zero-shot and fully-supervised results. As evaluation metrics, we follow the evaluation protocol developed by Maaz et al. (2023), employing GPT-3.5-turbo (Brown et al., 2020) to assess both accuracy and answer quality score.

### 4.2 QUANTITATIVE RESULTS

We present our long video action classification results in Table 1 for LVU (Wu & Krahenbuhl, 2021), Table 2 for Breakfast (Kuehne et al., 2014) and COIN (Tang et al., 2019), and compare HERMES 's performance against transformer-based models, including Object Transformers (Wu & Krahenbuhl, 2021), Movies2Scenes (Chen et al., 2023), and FACT (Lu & Elhamifar, 2024); hybrid state-space and transformer-based models such as Vis4mer (Islam & Bertasius, 2022), TranS4mer (Islam et al., 2023), and S5 (Wang et al., 2023); as well as the LLM-based model MA-LMM (He et al., 2024). For the MovieChat-1k dataset (Song et al., 2024), our results are presented in Table 3, where we compare against recent LLM-based models including MovieChat (Song et al., 2024), Video-ChatGPT (Maaz et al., 2023), Video-LLaMA (Zhang et al., 2023), and VideoChat (Li et al., 2023b). Our method achieves state-of-the-art performance across all datasets, with notable accuracy improvements of 7.3% on LVU and 14.9% on MovieChat-1k, significantly surpassing previous methods.

Table 1: **SOTA Comparison on the LVU Dataset:** The table presents Top-1 accuracy for various models. Unlike the minor incremental improvements observed among other methods, our model demonstrates a significant performance leap, outperforming its nearest competitor by 7.3%. The highest score is highlighted in **bold**, and the second highest is underlined.

| Model | Content | | | Metadata | | | | Avg |
|---|---|---|---|---|---|---|---|---|
| | Relation | Speak | Scene | Director | Genre | Writer | Year | |
| Object Transformer | 53.1 | 39.4 | 56.9 | 52.1 | 54.6 | 34.5 | 39.1 | 47.1 |
| VIS4mer | 57.1 | 40.8 | 67.4 | 62.6 | 54.7 | 48.8 | 44.8 | 53.7 |
| TranS4mer | 59.5 | 39.2 | 70.9 | 63.9 | 55.9 | 46.9 | 45.5 | 54.5 |
| S5 | 67.1 | 42.1 | 73.5 | 67.3 | 65.4 | 51.3 | 48.0 | 59.2 |
| Movies2Scenes | **71.2** | 42.2 | 68.2 | 70.9 | 57.8 | 55.9 | 53.7 | 60.0 |
| MA-LMM | 58.2 | 44.8 | 80.3 | 74.6 | 61.0 | 70.4 | 51.9 | 63.0 |
| **HERMES (Ours)** | 67.6 | **47.5** | **90.0** | **82.6** | **69.5** | **77.2** | **57.7** | **70.3** |

Table 2: **Performance comparison on Breakfast and COIN datasets** (Top-1 accuracy). Our method outperforms state-of-the-art models on both datasets.

| Model | Breakfast | COIN |
|---|---|---|
| FACT | 86.1 | - |
| VIS4mer | 88.2 | 88.4 |
| MA-LMM | 93.0 | 93.2 |
| S5 | 90.7 | 90.8 |
| TranS4mer | 90.3 | 89.2 |
| **HERMES (Ours)** | **95.2** | **93.5** |

Table 3: **Zero-shot performance on MovieChat-1k**. Our model significantly outperforms existing methods. The model marked with ‡ is fully supervised.

| Model | Global | | Breakpoint | |
|---|---|---|---|---|
| | Acc. | Score | Acc. | Score |
| MovieChat | 63.7 | 3.15 | 48.1 | 2.46 |
| Video-ChatGPT | 58.7 | 2.89 | 47.8 | 2.43 |
| Video-LLaMA | 56.3 | 2.72 | 45.8 | 2.11 |
| VideoChat | 60.2 | 3.08 | 46.3 | 2.32 |
| **HERMES (Ours)** | **78.6** | **4.23** | **57.3** | **3.29** |
| *HERMES (Ours)‡* | *84.9* | *4.40* | *65.8* | *3.65* |

## 4.3 ABLATION STUDIES

Ablations are conducted on the MovieChat-1k test set (global mode) using the zero-shot setting with additional ablations on the Breakfast dataset using the fully-supervised setting. These experiments focus on our two primary contributions, ECO and SeTR. For an extended and more comprehensive ablations, please refer to Appendix A.4.

**How important is ECO?** In Table 4, we demonstrate the critical role of ECO through several experiments. The results clearly indicate that the absence of our ECO and the Episodic Q-Former leads to a significant degradation in model performance due to the model lacking micro-level continuous representations. We further explore alternative update strategies, including randomly selecting features to retain (Rand.) and employing a first-in-first-out (FIFO) streaming approach. Our proposed update strategy outperforms both the Rand. and FIFO methods, highlighting its efficacy in retaining more relevant episodes. It is worth noting that during these ablations, SeTR remains active.

**How important is SeTR?** SeTR is designed to complement the episodic knowledge of our model with semantic insights. In Table 5, we observe that removing SeTR results in a 5% drop in accuracy. Additionally, we show that naive methods such as max pooling and average pooling are not as effective.

**Do we need a *hierachical* Q-Former?** Yes. We conducted an ablation study on the Breakfast dataset (Kuehne et al., 2014), to evaluate the efficacy of our proposed hierarchical Q-Former architecture. As shown in Table 6, our hierarchical Q-Former achieves superior performance with an accuracy of 95.2%, outperforming both flat frame-level ($fQFormer$, 93.2%) and video-level ($vQFormer$, 94.1%) architectures. This improvement can be attributed to the hierarchical structure's ability to capture multi-scale features, effectively aggregating information from frame to video level. By first processing frame-level details and then aggregating them at the video level, our ap-

Table 4: Ablations on the memory update design of our Episodic COmpressor (ECO).

|  | Acc. | Score |
|---|---|---|
| w/o | 55.1 | 3.55 |
| Rand. | 76.9 | 4.13 |
| FIFO | 77.1 | 4.15 |
| **ECO** | **78.6** | **4.23** |

Table 5: Ablations on different semantic compression methods.

|  | Acc. | Score |
|---|---|---|
| w/o | 73.3 | 4.09 |
| MaxPool | 70.4 | 3.99 |
| AvgPool | 73.3 | 4.04 |
| **SeTR** | **78.6** | **4.23** |

Table 6: Performance comparison between frame Q-Former, video Q-Former and our hierarchical Q-Former architecture.

|  | Acc. |
|---|---|
| Flat $fQFormer$ | 93.2 |
| Flat $vQFormer$ | 94.1 |
| Hierarchical $QFormer$ | **95.2** |

Table 7: Zero-shot performance comparison of MA-LMM with and without ECO and SeTR integration on MovieChat-1k.

| Model | Acc. | Score | Latency (s) |
|---|---|---|---|
| MA-LMM | 73.3 | 4.05 | 467 |
| MA-LMM + ECO | 76.7 (+3.4) | 4.14 (+0.09) | 266 (-43%) |
| MA-LMM + SeTR | 77.1 (+3.8) | 4.16 (+0.11) | 474 (+1.5%) |
| **HERMES (Ours)** | **78.6** (+5.3) | **4.23** (+0.18) | **250** (-46%) |

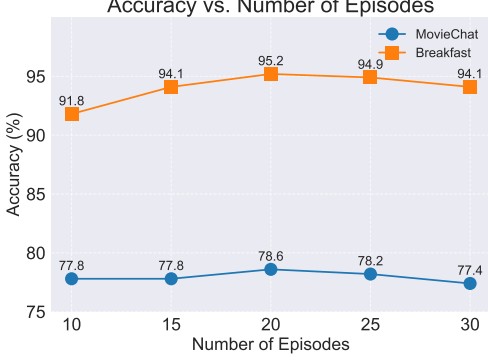

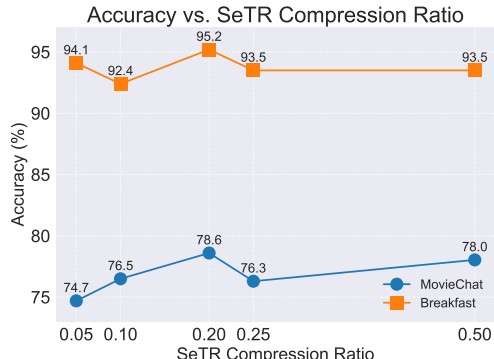

Figure 3: Effect of the number of ECO episodes on the model's accuracy on the MovieChat-1k and Breakfast datasets.

Figure 4: Effect of the SeTR 's keep ratio on the model's accuracy on the MovieChat-1k and Breakfast datasets.

proach mitigates information loss that may occur in direct video-level processing while avoiding the computational intensity of processing every frame individually.

**ECO as off-the-shelf memory manager.** We evaluate the impact of our Episodic COmpressor (ECO) as plug-ins to the existing MA-LMM model (He et al., 2024) by replacing the memory bank of MA-LMM. ECO is designed to efficiently process long video sequences while preserving temporal coherence and narrative structure. The results in Table 7 demonstrate substantial improvements with ECO integration with an accuracy increase by 3.4%. Moreover, ECO demonstrates superior efficiency compared to MA-LMM's memory bank, almost halving the overall inference latency.

**SeTR as off-the-shelf semantics retriever.** We also integrate our Semantics reTRiever (SeTR) into MA-LMM. SeTR is designed to enhance long video understanding by distilling high-level semantic cues, providing a cohesive framework for comprehending context and themes in long-form videos. As shown in Table 7, the integration of SeTR results in significant performance improvements. Accuracy increases by 3.8%, while the score improved by 0.11 points. Remarkably, these substantial performance gains were achieved with only a marginal 1.5% increase in inference time, highlighting the computational efficiency of SeTR and its potential for seamless integration into models requiring enhanced semantic representations.

Both SeTR and ECO yield significant performance enhancements when integrated with MA-LMM. SeTR showed a marginally higher performance boost in accuracy, which is expected given its role as a semantic add-on to MA-LMM's existing memory management system.

**How effective and efficient is ECO compared to other memory compressors?** To demonstrate the effectiveness and efficiency of our proposed ECO, we conduct a comparative analysis against two strong existing compression techniques: ToMe (Bolya et al., 2022) and MA-LMM (He et al., 2024) in Figure 5. We calculate the inference time for each model on the MovieChat-1k dataset. Powered by ECO our model achieves the highest accuracy (78.6%) among all models, outperforming MA-LMM by 5.3% and ToMe by a substantial 13.8%. HERMES achieves the highest inference speed among the compared models, while also maintaining superior accuracy. It is slightly faster than ToMe and significantly outperforms MA-LMM, reducing inference time by 46% com-

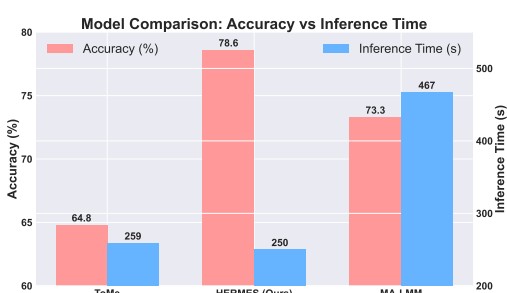

Figure 5: Our method is 46% faster than MA-LMM while being 5.3% more accurate, and registers an absolute gain of 13.8% accuracy compared to ToMe.

pared to the latter. These results demonstrate our model's ability to deliver state-of-the-art accuracy without compromising on efficiency.

**Hyperparameters for ECO and SeTR.** Our experiments on the MovieChat-1k (zero-shot) and Breakfast (fully-supervised) datasets reveal compelling insights into the optimal configuration of ECO (Figure 3) and SeTR (Figure 4). For ECO, we discover that an episodic memory size of 20 consistently yields peak performance across both datasets, achieving a 78.6% accuracy on MovieChat-1k and a 95.2% on Breakfast. This sweet spot balances comprehensive video representation with computational efficiency, as larger memory sizes show diminishing returns. SeTR's performance proved equally intriguing, with a keep ratio of 20% (reducing representations by 80%) emerging as the optimal choice. Remarkably, even at extreme keep ratios as low as 5%, our model exhibits robust performance, with a very slight decrease in accuracy. Such results demonstrate the resilience of *HERMES* to hyperparameter variations suggesting that it is suitable for deployment across diverse video understanding datasets with minimal hyperparameter tuning.

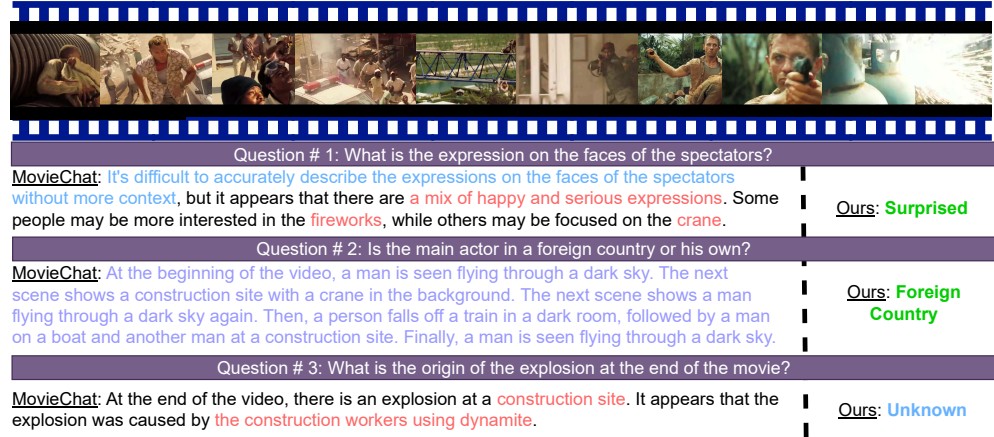

Figure 6: **Qualitative Results:** We select a challenging video from the MovieChat-1k dataset and pose various difficult questions to both MovieChat (Song et al., 2024) and HERMES. The results demonstrate our model's superior ability to answer both fine-grained questions (Q1 and Q3) and general questions (Q2). Answers highlighted in blue denote tentative answers, red denote wrong answers, purple denote hallucinations, and green denote correct answers.

### 4.4 QUALITATIVE RESULTS

We present qualitative results on a challenging movie scene from the MovieChat-1k dataset to evaluate our model's capability in answering both fine-grained and general questions about an extended video (14k frames). To rigorously assess the models, we bypass the original Q&As from the dataset (e.g., Q: What's the time in the video? A: Day, ...) and ask questions that require a deeper understanding of the scene. Our model accurately responds to these questions while exhibiting a candid acknowledgment of its limitations (e.g., Q3). In contrast, MovieChat (Song et al., 2024) frequently generates hallucinated and incorrect answers. For instance, in response to Q2, MovieChat avoids answering the question. Our model achieves this performance by processing only 100 out of the 14k frames (approximately 0.7%), whereas MovieChat processes 2,048 frames, more than 20 times the data utilized by HERMES. For failure cases of our model, please refer to Appendix A.5.

## 5 RELATED WORK

**Action recognition** is an essential task in video understanding, primarily focusing on identifying specific actions within short video clips. Various approaches have been developed, with convolutional neural networks forming the core of many of them. Early work by Ji et al. (2012) utilized 3D convolutions, while Varol et al. (2017) employed temporal convolutions. 2D CNNs coupled with temporal modeling have also been explored, with representative works such as Temporal Difference Networks (TDN) (Ng & Davis, 2018) and Event Adaptive Networks (EAN) (Tian et al., 2022). More recently, transformer-based models have gained prominence with works such as Faure et al. (2023), Xu et al. (2021), and Zhang et al. (2022).

**Video question answering (VideoQA)** aims to answer questions related to video content, requiring a deep understanding of both visual and textual information. Datasets such as ActivityNet-QA (Yu et al., 2019) for short videos, and MovieChat-1k for long videos (Song et al., 2024) provide benchmarks for evaluating models in this field, allowing for several research endeavors on this subject (Zhang et al., 2020; Zhuang et al., 2020; Pan et al., 2023).

**Long-form video understanding** presents unique challenges due to the extended duration and complex narrative structures involved. Datasets with these properties include LVU (Wu & Krahenbuhl, 2021), COIN (Tang et al., 2019), Breakfast (Kuehne et al., 2014), and MovieChat-1k (Song et al., 2024). Traditional approaches to tackling such a task often extend methods designed for short videos to handle longer sequences, such as pooling over the temporal dimension (Tang et al., 2020; Faure et al., 2023). Other methods such as Wu & Krahenbuhl (2021); He et al. (2024); Wu et al. (2022) and Song et al. (2024) explore memory techniques via token compression. Additionally, Tian et al. (2024) introduced a video semantic compression framework using low-level bitrate coding. Wang et al. (2023) introduced selective structured state-spaces for long-form videos, followed by others Islam & Bertasius (2022); Islam et al. (2023) exploiting the ability of state-space models to retain long-term context.

**LLM-based Long-Form Video Understanding:** Recent advancements in large language models (LLMs) (Touvron et al., 2023; Chiang et al., 2023) have piqued researchers' curiosity regarding their use for video understanding (Maaz et al., 2023). It turns out to be a good match, as understanding videos often involves transforming their content into words, whether it's video captioning, video question answering, or even action classification. Song et al. (2024) and He et al. (2024) propose frameworks that employ memory techniques to handle extensive video content while Ren et al. (2024) presents TimeChat, explicitly conditioning the model to manage time-dependent information.

## 6 CONCLUSION

We propose **HERMES**, a novel framework designed to enhance long-form video understanding through two key components inspired by cognitive processes. The first, Episodic COmpressor (ECO), captures representations as sequences of continuous actions, reflecting episodic memory. The second, Semantics reTRiever (SeTR), serves as a high-level summarizer, distilling essential semantic information. Our model achieves state-of-the-art results on several long-video datasets, significantly outperforming existing methods. Through experiments on LVU, Breakfast, COIN and MovieChat, we have demonstrated the effectiveness and efficiency of ECO and SeTR.

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

# A APPENDIX

The Appendix is organized as follows:

- **A.1** Reproducibility Statement
- **A.2** Implementation Details
- **A.3** Model Details
- **A.4** Extended Ablations
- **A.5** More Qualitative Results
- **A.6** Error Analysis: When does HERMES fail and why?
- **A.7** How is our approach related to cognitive processes?

## A.1 REPRODUCIBILITY STATEMENT

To facilitate the reproducibility of our work, we will make our code, pretrained models, default hyperparameters, and preprocessed annotations publicly available. Detailed hyperparameters for each dataset are also provided in Table 8. Our model demonstrates efficient performance, completing inference on the MovieChat-1k test set in 13 minutes (22 FPS) using a single V100 GPU (32 GB), and training on the MovieChat-1k dataset in less than 12 minutes with 8x 32 GB GPUs. In contrast to recent LLM-based approaches that necessitate extensive and costly multi-stage pretraining on increasingly large datasets, our model is designed for accessibility, thereby lowering the barrier for researchers without access to high-end computing resources. We achieve high performance while maintaining accessibility by leveraging existing pretrained weights and implementing our training-free ECO and SeTR, resulting in a model where finetuning is optional.

## A.2 IMPLEMENTATION DETAILS

To ensure the reproducibility of our results, we provide the training details, which are also the defaults in our soon-to-be-released code. These settings are mostly consistent across different datasets.

Table 8: Hyperparameters used for different datasets.

| Dataset | Max Epochs | LR | Batch | Frames (N) | Episodes | Keep Ratio |
|---|---|---|---|---|---|---|
| MovieChat-1k (G) | 1 | 1e-4 | 32 | 100 | 20 | 0.2 |
| MovieChat-1k (B) | 1 | 1e-4 | 32 | 40 | 10 | 0.5 |
| LVU | 20 | 1e-4 | 32 | 100 | 20 | 0.2 |
| COIN | 20 | 1e-4 | 32 | 100 | 20 | 0.2 |
| Breakfast | 20 | 1e-4 | 32 | 100 | 20 | 0.2 |

LR is the learning rate, and Keep Ratio is the SeTR keep ratio. Episodes refer to the number of episodes to which we compress the input frames (i.e., the capacity of ECO). The number of frames (N) represents the quantity of frames retained from the original video to serve as input to the model. These frames are selected by applying a regular stride over the original video's frame sequence, where the stride length is determined by the ratio of original frame count to N. *Max Epoch = 20* means we run the program for 20 epochs, performing evaluation after each epoch, and then pick the model with the highest validation accuracy. MovieChat-1k (G) and MovieChat-1k (B) denote global and breakpoint modes, respectively. All models were trained on 8 V100 GPUs (32GB VRAM each).

## A.3 MODEL DETAILS

### A.3.1 DETAILS OF OUR EPISODIC QFORMER

The Episodic Q-Former, as visualized in Figure 7, extends the original QFormer architecture by inserting the Episodic COmpressor (ECO) described in Section 3.2. It begins with a set of initial queries that undergo a self-attention process, enhancing internal query representations. These

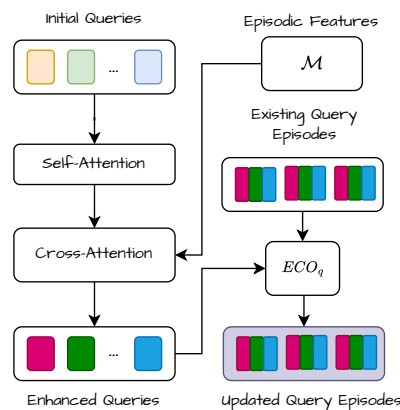

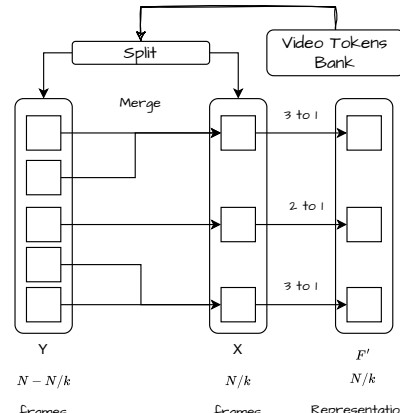

Figure 7: **Illustration of our Episodic QFormer:** We insert our ECO in the original QFormer to effectively and efficiently compute and aggregate queries across long video sequences. It returns query episodes representing the whole video.

Figure 8: **Illustration of SeTR:** Our Semantics reTRiever uses a stride of $k$ split the videos into groups $X$ of $N/k$ frames and $Y$ of $N - \frac{N}{k}$ frames, then merge each frame from $Y$ to its most semantically similar in $X$.

queries then interact with episodic visual features through cross-attention, allowing the incorporation of contextual visual information. The resulting enhanced queries are fed into our ECO module alongside existing query episodes, which represent previously processed queries grouped into episodes. ECO iteratively updates the query episodes, adding the new queries to the existing episodes. This Episodic QFormer allows the model to better handle long sequences or repeated queries by maintaining richer contextual knowledge across iterations.

### A.3.2 DETAILS OF SETR

We design SeTR as an efficient tool to retrieve semantic information from a long video. Given tokens extracted from a long video sequence, we use a stride of size $k$, to form a group of $\frac{N}{k}$ frames representing the number of semantics we want to extract. We then compress the remaining $N - \frac{N}{k}$ frames into extracted $\frac{N}{k}$ frames to obtain the semantic representations. SeTR is illustrated in Figure 8.

### A.4 EXTENDED ABLATIONS

#### A.4.1 HOW DOES THE NUMBER OF FRAMES AFFECT THE MODEL'S ACCURACY AND LATENCY?

MovieChat (Song et al., 2024) processes 2048 frames for each video, while we use only 100 frames, as previous studies have demonstrated how redundant video data is (Simonyan & Zisserman, 2014; Wang et al., 2016). Given that the MovieChat-1k dataset contains very long videos (some exceeding 14,000 frames), we conducted experiments to extend the number of frames our model processes. Specifically, we experiment with 40, 80, 100, 300, 500, and 1000 frames while keeping the number of episodes constant. As for the SeTR keep ratio, we decrease it in function of the number frames so that the number of semantic features we keep equals 20.

We observe a complex relationship between model accuracy, processing latency, and the number of frames analyzed. Figure 9 illustrates these relationships, providing insights into the performance trade-offs of our model. As evident from Figure 9, the relationship between accuracy and the number of frames is non-monotonic. Accuracy initially increases as the number of frames grows, reaching

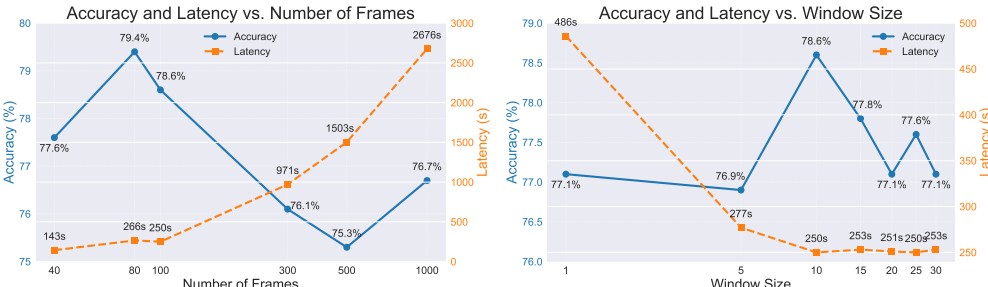

Figure 9: **Accuracy and latency as functions of the number of frames processed:** This figure demonstrates the non-monotonic relationship between accuracy and frame count, with peak performance at 80 frames. Latency increases super-linearly with frame count while accuracy stalls, highlighting the redundancy of video data.

Figure 10: **Accuracy and latency as functions of input window size:** The graph illustrates the interplay between model accuracy, processing latency, and the window size. Notably, accuracy peaks at a window size of 10, while latency stabilizes for window sizes of 10 and above. In all cases the accuracy only slightly fluctuates.

a peak of 79.4% at 80 frames with a modest latency (note that we use 100 frames as the default parameter in other experiments for consistency with other datasets). This suggests that up to this point, additional frames provide valuable context that enhances the model's understanding. However, beyond 80 frames, we observe a decline in accuracy, possibly due to the introduction of noise or irrelevant information from temporally distant parts of the video.

Latency, on the other hand, exhibits a near-linear increase with the number of frames up to 300 frames, after which it grows super-linearly. This rapid increase in latency for higher frame counts underscores the computational challenges of processing large numbers of frames, particularly in real-time or near-real-time applications.

Interestingly, the model's performance at 1000 frames (76.7% accuracy) is lower than its performance at 40 frames (77.6% accuracy), but with a significantly higher latency (2676s vs. 143s). This observation highlights the diminishing returns and potential drawbacks of simply increasing the number of processed frames. It also underscores the importance of thoughtful frame selection in video understanding tasks. Future work could explore adaptive frame selection techniques that dynamically adjust the number of frames based on video content, potentially optimizing both accuracy and efficiency.

### A.4.2 How does the window size affect the model's accuracy and latency?

Our analysis of our model's zero-shot performance on the MovieChat-1k test set reveals intriguing relationships between accuracy, latency, and input window size. Figure 10 illustrates these trade-offs. As evident from Figure 10, the relationship between accuracy and window size is non-monotonic. Accuracy initially increases with window size, reaching a peak of 78.6% at a window size of 10. This suggests that providing more context to the model improves its performance up to a certain point. However, beyond this optimal window size, accuracy begins to decline, possibly due to the introduction of irrelevant context.

Latency exhibits a sharp decrease from window size 1 to 5, after which it remains relatively stable. This indicates that while smaller window sizes may seem computationally advantageous, they incur higher latency, possibly due to the need for more frequent ECO call. The optimal trade-off occurs at a window size of 10, where we observe peak accuracy and stabilized latency suggesting that carefully tuned context windows can enhance long-form video understanding without incurring additional computational costs.

### A.4.3 A NOTE ON LATENCY

The MovieChat-1k test set comprises 170 videos, from each of which our model samples 100 frames. This results in a total of 17,000 frames to be processed. Our empirical measurements show that the model requires 774 seconds to complete end-to-end inference on this dataset using a single V100 GPUs (32GB VRAM). This translates to a processing speed of approximately **22 frames per second (FPS)**, which is very close to real-time performance. Such a result suggests that our approach is not only effective in terms of accuracy but also efficient enough for practical applications in video understanding tasks.

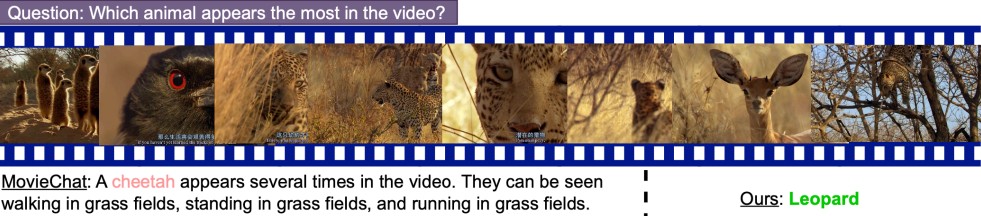

(a) **Animal Identification:** MovieChat mistakenly identifies a Leopard as a Cheetah, even though no Cheetah appears in the video.

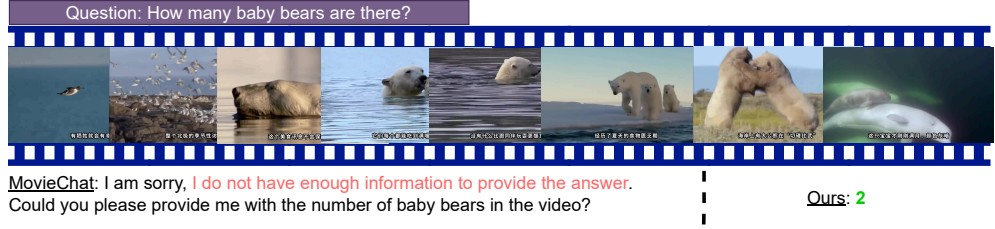

(b) **Animal Counting:** This question is particularly challenging because the bears appear infrequently in the video, and the question specifically asks about "baby bears." Despite MovieChat analyzing 2048 frames and our model only analyzing 100 frames, our model was able to locate and count the baby bears accurately.

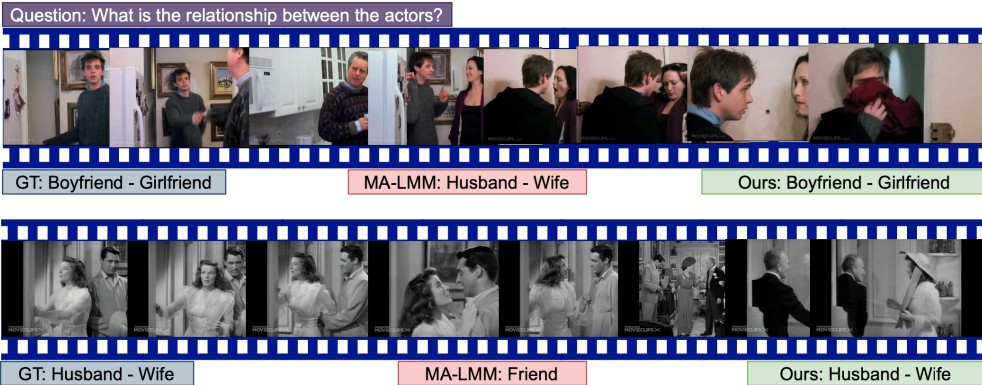

(c) **Determining People's Relationships:** We compare our results with those of MA-LMM, with both models trained on the LVU dataset. Thanks to our episodic memory compression, our model excels at determining people's relationships across thousands of frames of interactions.

Figure 11: Qualitative results demonstrating the capabilities of our model compared to MovieChat and MA-LMM across different tasks. (a) Animal identification shows MovieChat's confusion between Leopard and Cheetah. (b) Animal counting highlights the challenge of locating baby bears with limited appearances in the video, where our model outperforms despite fewer frames. (c) Relationship determination benefits from our episodic memory compression, enabling better identification of relationships over extended interactions.

## A.5 MORE QUALITATIVE RESULTS

To further illustrate the capabilities of our model, we present a series of qualitative examples that highlight its strengths in various long-form video understanding tasks.

**Animal Identification.** Figure 11a demonstrates our model's superior performance in animal identification compared to MovieChat. In this example, MovieChat incorrectly identifies a leopard as a cheetah, despite no cheetah being present in the video. This misidentification underscores the importance of accurate visual feature extraction and semantic understanding in long-form video analysis.

**Animal Counting.** Figure 11b showcases our model's ability to perform complex counting tasks, even with limited information. The task involves counting baby bears, which appear infrequently in the video. Despite analyzing only 100 frames compared to MovieChat's 2048 frames, our model accurately locates and counts the baby bears. This demonstrates the efficiency of our ECO and SeTR modules in capturing and retaining crucial information from sparse appearances.

**Determining People's Relationships.** In Figure 11c, we compare our model's performance against MA-LMM in determining relationships between people over extended video sequences. Both models were trained on the LVU dataset. Our model's superior performance in this task can be attributed to the episodic memory compression technique, which allows for better retention and analysis of interactions across thousands of frames.

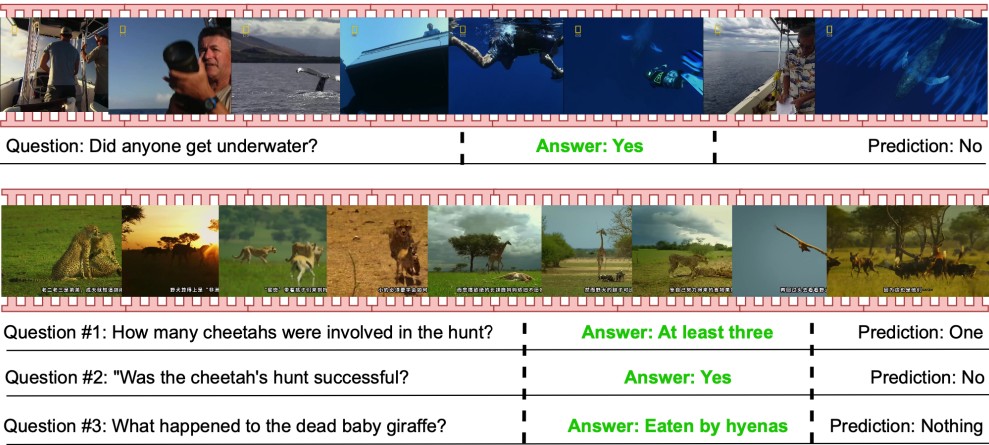

Figure 12: **Where and when HERMES fail:** The top row shows a marine life video where the model fails to recognize underwater scenes. The bottom row depicts a wildlife documentary where the model struggles with quantitative reasoning and event inference across multiple frames. These cases highlight limitations in contextual understanding and temporal information integration.

### A.5.1 VISUALIZATION OF ECO AND SeTR

Figure 13 demonstrates the inner-workings of ECO and SeTR. The top row illustrates a curated summary of the video content, highlighting diverse scenes, such as landscapes, wildlife, and environmental features.

SeTR is responsible for extracting high-level semantic features and grouping frames with similar themes, as shown in the mid row. For instance, the module effectively captures thematic clusters such as "Landscape," "Various Birds," and "Reptiles," providing a concise overview of the video.

Meanwhile, ECO processes the video at a more granular level, segmenting it into coherent episodes that reflect the narrative flow. The bottom row showcases this segmentation, organizing the content into episodic units like "Arid Landscape," "Lake and Aquatic Bird," and "Flies." This two-tiered approach ensures both thematic abstraction and temporal coherence, enabling a comprehensive understanding of the video.

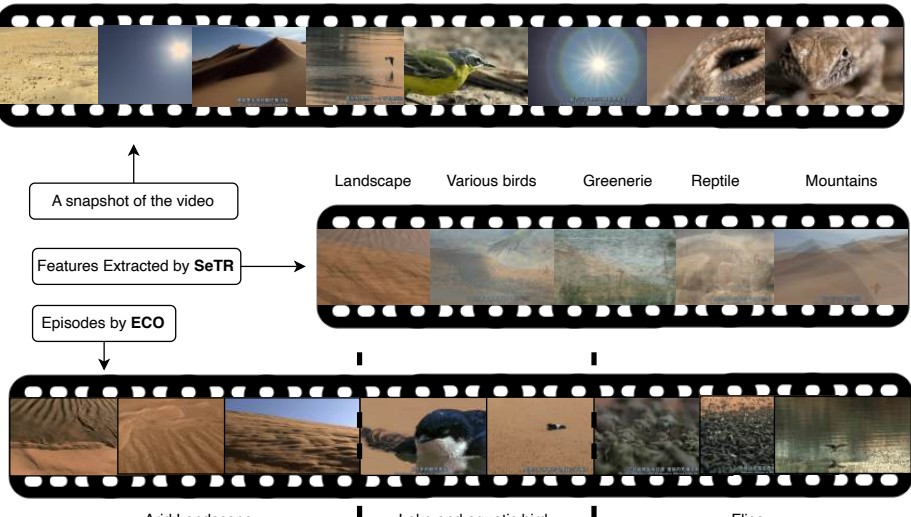

Figure 13: **Visualization of ECO and SeTR:** The top row presents a curated visual summary of the video, showcasing key scenes such as landscapes, wildlife, and environmental features. The middle row highlights the functionality of SeTR, which extracts semantic features and clusters frames into thematic groups, including "Landscape," "Various Birds," and "Reptiles." Finally, the bottom row illustrates the operation of ECO, which segments the video into coherent narrative episodes, such as "Arid Landscape," "Lake and Aquatic Bird," and "Flies." Together, these modules provide both high-level abstraction and detailed episodic structure for comprehensive video understanding.

## A.6 ERROR ANALYSIS: WHEN DOES HERMES FAIL AND WHY?

Our model, while generally effective, demonstrates several notable failure cases that warrant further investigation and improvement. Figure 12 illustrates examples where the model's predictions deviate from ground truth answers, revealing key limitations in contextual reasoning and temporal information integration. Figure 12 presents two sets of video frame sequences that highlight shortcomings in our model's performance. In the top row, we observe a documentary on marine life. Despite clear visual cues of underwater scenes and diving equipment, the model incorrectly predicts that no one got underwater. The bottom row showcases a more complex scenario from a wildlife documentary. Here, the model exhibits multiple errors: It underestimates the number of cheetahs involved in the hunt, predicting only one when at least three are present. This indicates a weakness in quantitative reasoning across temporally distributed information. The model incorrectly predicts that the cheetah's hunt was unsuccessful, contradicting the visual evidence. This error points to difficulties in inferring outcomes from sequences of events. Lastly, the model fails to recognize the fate of a dead baby giraffe, predicting "nothing" when the correct answer is "eaten by hyenas".

These examples emphasize the need for improved mechanisms to aggregate and reason over long-range temporal dependencies, as well as enhanced capabilities in scene understanding and event inference.

## A.7 HOW IS OUR APPROACH RELATED TO COGNITIVE PROCESSES?

Our approach to long-form video understanding is inspired by cognitive processes involving memory and comprehension. According to the literature on neuroscience (Tulving et al., 1972; Schacter & Tulving, 1982; Tulving, 1983), human cognition involves two primary types of memory: episodic and semantic. Episodic memory is the ability to recall specific events or episodes, while semantic memory refers to the storage of general knowledge and concepts. These forms of memory are crucial for understanding long-form narratives, where a coherent understanding arises from the integration of specific events and overarching themes.

The proposed HERMES model incorporates these cognitive processes through its two main components, ECO and SeTR. ECO, akin to the function of episodic memory, selectively retains and compresses key events from the video, allowing the model to form a structured representation of the narrative as it unfolds. This approach is an oversimplified abstraction of findings in cognitive neuroscience, which highlight the role of the hippocampus in the consolidation of episodic memories (Eichenbaum, 2004; Schacter & Tulving, 1982), and the concept of *subjective time* (Arstila et al., 2014) that sees a scene (or a video) not as a series of frames but as a series of experiences. The hippocampus enables the organization of temporally distinct experiences into a coherent memory trace, something that we aim to capture with ECO. Moreover, the sequential processing and aggregation of information in our model align with the concept of event segmentation in cognitive psychology (Zacks et al., 2007). Humans naturally segment continuous experiences into discrete events, which aids in memory formation and recall.

Meanwhile, SeTR functions similarly to semantic memory, extracting and reinforcing high-level semantic cues. This process mirrors how the brain integrates detailed episodic memories with broader semantic knowledge stored in the neocortex (McClelland et al., 1995; Binder & Desai, 2011). Also related is the concept of gist extraction which involves rapidly comprehending the essence or overall meaning of a scene or situation (Oliva, 2005). This ability allows humans to quickly understand the context of a complex scene without processing every detail. Our SeTR operates similarly by identifying and extracting high-level semantic cues that provide a concise overview of the scene and actions.

The integration of these cognitive processes not only aligns with human-like comprehension but also offers a framework for efficiently handling the vast and diverse information present in long-form videos. Significant improvements over existing state-of-the-art models, underscore the effectiveness of this cognition-inspired approach. While our model is a oversimplified abstraction of human cognition, it provides a foundation for exploring more complex cognitive mechanisms in future work.

