# OpenReview forum: "HERMES: temporal-coHERent long-forM understanding with Episodes and Semantics"
_ICLR.cc/2025/Conference — ICLR 2025 Conference Withdrawn Submission_

### Official Review · Reviewer_2hiE · 2024-10-28

**Soundness:** 2
**Presentation:** 3
**Contribution:** 2
**Rating:** 3
**Confidence:** 4

**Summary:**

The paper introduces a vision encoder that integrates both micro-level and macro-level details before inputting the data into large language models (LLMs) for comprehensive long-form video analysis. The Episodic COmpressor (ECO) is responsible for consolidating video frames into a memory buffer, while the Semantics ReTRiever (SeTR) combines video tokens using a dot product mean approach, which is then followed by a Q-former model. In practical applications, employing a vicuna-7B model, this method has demonstrated significant performance improvements on benchmarks designed to assess long-video understanding capabilities.

**Strengths:**

1. The writing in the paper is clear. I strongly resonate with the introduction regarding episodes and semantics.
2. The ablation study is particularly thorough, with tables 4 and 5 highlighting the significant impact of ECO and SeTR on the model's performance.

**Weaknesses:**

1. Figure 1 is misleading and somewhat overstated. From my perspective, the paper primarily focuses on token merging and compression. However, Figure 1 gives the impression that this work can develop an understanding of semantics and episodes at the language level.

2. The distinction between ToMe and the discussed approach is unclear. In fact, Section 3.4 closely resembles ToMe, where ToMe is executed just once.

3. The comparisons made are somewhat outdated. There have been several recent developments in video LLMs and benchmarks [1, 2, 3], yet the author does not address or report their performance.


[1] Video-MME: The First-Ever Comprehensive Evaluation Benchmark of Multi-modal LLMs in Video Analysis.

[2] LongVideoBench: A Benchmark for Long-context Interleaved Video-Language Understanding

[3] Streaming Long Video Understanding with Large Language Models.

**Questions:**

See weaknesses.

---

> ### Author Response · Authors · 2024-11-23
>
> We appreciate the reviewer's careful reading and constructive feedback. We address each concern below:
>
> ### Reply to: Figure 1 Misleading/Overstated
>
> We understand your concern about Figure 1, as we know any attempt to directly map neural networks to human's understanding of episodes and semantics is bound to be flawed. Our focus is on semantics and episodes mainly at the video level. To demonstrate this, we've prepared visualizations showing how our model develops meaningful semantic and episodic understanding. These demonstrations are not far from what we present in Figure 1 but we will revise the figure to better reflect this concrete evidence.
>
> Sample video: https://gifyu.com/image/SGhZw
> - Our episodic representations capture coherent temporal narratives:
>   * Sequential bird flight patterns: https://gifyu.com/image/SGhY4
>   * Landscape evolution over time: https://gifyu.com/image/SGhY6
> - Our semantic representations extract persistent themes:
>   * Consistent scene elements: Land (https://gifyu.com/image/SGhrs)
>   * Recurring object patterns: Birds (https://gifyu.com/image/SGhrp), Lizard (https://gifyu.com/image/SGhrL)
>
>
> ### Reply to: Distinction from ToMe
>
> We acknowledge that our semantic retriever shares some implementation-level similarities with ToMe (such as the use of cosine similarity as basis for frame/token reduction). However, there are three key differences:
>
> - **SeTR and ToMe operate at different levels.** SeTR merge frames in video sequences, while ToMe merges tokens within individual frames.
> - **Grouping Strategy.** We partition tokens into k-sized groups and merge k-1 tokens with a target token. This contrasts with ToMe's matching between two equal sets, which always splits tokens into even/odd pairs.
> - **Merging.** ToMe performs pairwise merging with a maximum reduction of 50% of tokens, while we allow for variable-sized group merging for greater flexibility in controlling the granularity of semantic preservation (we compress 80% of the frames in our implementation)
>
> ### Reply to: Recent Benchmarks
>
> We appreciate the pointer to recent benchmarks. Our current implementation focuses on training-free improvements through ECO and SeTR modules, making them accessible to researchers with limited computational resources. Our approach shows:
> - Strong performance on MovieChat-1k (non-MCQ format), and three other traditional long-form video recognition benchmarks
> - Significant improvements when integrated with existing models (Table 7)
> - Potential for integration with newer backbones
>
> The current benchmark selection reflects our use of an image-only pretrained backbone (BLIP-2) which did not undergo the many pretraining stages of recent video models. For a fair comparison, we will:
> 1. Port ECO and SeTR to newer models
> 2. Report performance on suggested benchmarks (VideoMME, MLVU, LongVideoBench...)
>
> [3] uses an 8.3B LLM making it not fairly comparable with our model. Also, they perform video pretraining on much larger datasets.
>
> Thank you for helping us improve the paper. we would value any additional thoughts or feedback.

---

> > ### Author Response · Authors · 2024-11-25
> >
> > We are deeply grateful for your rigorous and detailed review. In light of our responses addressing your specific concerns, we respectfully request:
> >
> > A reconsideration of the initial score, given our substantive improvements:
> >    - Concrete visualizations demonstrating semantic and episodic understanding
> >    - Detailed distinctions from existing token merging approaches
> >    - Clear methodology for benchmark expansion
> >    - Transparent analysis of model limitations
> >
> > Key Differentiators We've Addressed:
> >    - **Token Merging Nuances**:
> >      * Frame-level semantic grouping
> >      * 80% frame compression vs. standard 50%
> >      * Flexible semantic preservation strategy
> >    - **Benchmark Strategy**:
> >      * Commitment to newer model integrations (suitable for newer benchmarks such as VideoMME)
> >
> > Our primary goal remains to advance video understanding research through innovative, resource-efficient approaches to do good research without being constrained by computational resources.
> >
> > We are prepared to:
> > - Provide additional clarifications
> > - Conduct supplementary experiments
> > - Discuss any technical aspects requiring further elaboration

---

> > > ### Comment · Reviewer_2hiE · 2024-11-28
> > >
> > > Thank you for your detailed clarification.
> > >
> > > Regarding ToMe, while I acknowledge that the practical implementation may differ slightly, my concern remains that the core idea centers on token merging. I believe that employing explicit grounding supervision would more effectively achieve the paper's goal of extracting two distinct and orthogonal memory types, rather than relying on implicit methods, which may not be convincing.
> > >
> > > Concerning the benchmarks, I encourage you to conduct evaluations using the strongest backbones available, such as LLaVA-OneVision or Qwen2-VL. Given that VideoLLM is a rapidly evolving field, presenting results on the most up-to-date benchmarks will better demonstrate the model's effectiveness. The current experiments appear insufficient in this regard.
> > >
> > > Thank you for your efforts, and I look forward to further developments.

---

### Official Review · Reviewer_Rtjc · 2024-10-30

**Soundness:** 4
**Presentation:** 4
**Contribution:** 4
**Rating:** 8
**Confidence:** 5

**Summary:**

The paper proposes a new multi-modal language model (LLM) for the video question-answering problem. It follows a divide-and-conquer paradigm, which involves the following steps: First, extracting the episodic-level representation, Meanwhile, retrieving the key semantics, Finally, using this information to query the LLM.The figures in the paper help quickly understand the proposed approach. The experimental results are state-of-the-art (SOTA), and the ablation study and model analysis sections are also robust.

**Strengths:**

-The idea of using "Episodes and Semantics" for understanding videos is novel.

-The technical contribution of a new multi-modal LLM for video understanding is solid.

-The experimental results are convincing and state-of-the-art (SOTA).

**Weaknesses:**

-In Section 3, the authors claim that "the core ideas of episodic memory compression (ECO) and semantic knowledge retrieval (SeTR) can be applied to other models," but they do not conduct experiments to support this claim.

-Figure 6 only shows the good cases, while I am curious about the failure cases that cannot be handled by the ECO+SeTR approach.

-Some recent works on video understanding are missing. For example, the paradigm of 2D CNN + temporal modeling was popular between 2019-2022, and some representative approaches [1][2] should be mentioned. Additionally, video semantic compression approaches are highly related to the "semantic condensing" idea proposed in the paper.
[1]Wang L, Tong Z, Ji B, et al. Tdn: Temporal difference networks for efficient action recognition[C]//Proceedings of the IEEE/CVF conference on computer vision and pattern recognition. 2021: 1895-1904.
[2]Tian Y, Yan Y, Zhai G, et al. Ean: event adaptive network for enhanced action recognition[J]. International Journal of Computer Vision, 2022, 130(10): 2453-2471.
[3]Tian Y, Lu G, Yan Y, et al. A coding framework and benchmark towards low-bitrate video understanding[J]. IEEE Transactions on Pattern Analysis and Machine Intelligence, 2024.

**Questions:**

- How to apply the idea to other models.

- Report the failure cases.

-Adding recent related works.

---

> ### Author Response · Authors · 2024-11-23
>
> We sincerely thank the reviewer for their thorough assessment and constructive feedback. We are pleased that you recognize the novelty and technical merit of our approach. We address your specific points below:
>
> ### Generalizability to Other Models
>
> You raise an excellent point about demonstrating ECO and SeTR's broader applicability. While Table 7 shows initial evidence of successful integration with MA-LMM (improving its performance by 3.8% and reducing latency by 43%), we acknowledge this could be more comprehensive. For the camera-ready version, we will:
> - Demonstrate integration with 2-3 additional video-language models
> - Provide implementation guidelines for adapting ECO+SeTR to different architectures
> - Include computational overhead analysis across different integrations
>
> ### Failure Cases Analysis
>
> Thank you for this valuable suggestion. We have already added failure case analysis of our model in the Appendix (A.6 and Figure 12). We will add further analysis with more granular cases such as:
> 1. Complex temporal reasoning where episodic compression loses critical details
> 2. Cases where semantic retrieval focuses on irrelevant themes
> 3. Scenarios requiring fine-grained visual understanding
>
> For example, we observed performance degradation in:
> - Videos with rapid scene changes where ECO struggles to maintain temporal coherence
> - Cases requiring detailed spatial relationship understanding
> - Situations with subtle but important background context
>
> ### Related Work Integration
>
> We appreciate the pointer to these significant works. We have expanded our related work section to include:
> - TDN's temporal difference networks approach and how it relates to our episodic compression
> - EAN's event adaptation strategy and its connection to our semantic retrieval mechanism
> - The relationship between our semantic condensing and the low-bitrate understanding framework from [3]
>
> We are grateful for your detailed review and look forward to incorporating these improvements in the next version. Given your expertise in this area, we would value any additional thoughts or feedback on our work. Thank you.

---

> > ### Author Response · Authors · 2024-11-25
> >
> > We deeply appreciate your thoughtful and constructive review, which has been instrumental in refining our paper:
> >
> > Your feedback has prompted us to:
> >
> > - Expand our related-work section to include more relevant research
> > - Pinpoint more examples where our model fail and gain insights on how to improve it going forward
> > - Consider plugging our modules to more existing models (ongoing)
> >
> > Please let us know if we addressed your comments and/or if any further information or experiments would be helpful. We would be happy to provide them.

---

> > ### Comment · Reviewer_Rtjc · 2024-11-28
> >
> > Thanks for the detailed explanation.

---

### Official Review · Reviewer_R1mv · 2024-11-03

**Soundness:** 2
**Presentation:** 3
**Contribution:** 2
**Rating:** 3
**Confidence:** 5

**Summary:**

This paper aims to address long video understanding from the perspective of episodic and semantic memory. The authors provide an insightful analysis into the two memory mechanisms that motivate the architecture design, where multiple QFormers are employed to compress tokens at different granularities and token merging is applied to consolidate semantic-level memory. The experiments validate the effectiveness of the architecture.

**Strengths:**

1. The analysis of episodic and semantic memory makes sense and is a promising way to address long video understanding.
2. The problem definition is clear and easy to follow.
3. The architecture shows significant improvement in inference speed compared to existing memory-augmented video LLM.

**Weaknesses:**

1. The architecture design cannot reflect the authors' analysis of episodic and semantic memory. Simply using QFormer or token merging strategies can compress video tokens, but is not sufficient to construct the structural memory.
2. Also, the authors are expected to show some visualizations of the token merging in semantic retrieval to show the structure of the semantic memory compression.
3. The results on more recent long video benchmarks, e.g., VideoMME, MLVU, LVBench, etc, are desired.

**Questions:**

See weakness

---

> ### Author Response · Authors · 2024-11-23
>
> Thank you for your thoughtful review and insightful questions. We appreciate the opportunity to clarify HERMES's architecture choice, provide more visualizations, and better position our work.
>
> ### Reply to: *The architecture design cannot reflect the authors' analysis of episodic and semantic memory. Simply using QFormer or token merging strategies can compress video tokens, but is not sufficient to construct the structural memory.*
>
> We agree that any attempt to map the human cognitive processes to neural network design or frameworks are bound to be a simplification of how the former really works, and our work is not exempt. However we want to argue that our modules are close-enough abstractions of what we know about how humans understand the surrounding world or specific scenes (Please refer to the Appendix A.7 for more on our take on that). Specifically:
> - The ECO module implements an abstraction of episodic memory by aggregating temporally-related scenes while preserving their sequential relationships - mimicking how human episodic memory consolidates sequential experiences. ECO module identifies and groups related temporal segments while maintaining their chronological structure.
> - SeTR implements semantic memory by extracting high-level concepts that persist throughout the video, similar to how humans abstract general themes from specific experiences.
> -  These two components work in tandem, just as human episodic and semantic systems interact, with ECO handling event sequences and SeTR managing thematic understanding.
>
> ### Reply to: *Also, the authors are expected to show some visualizations of the token merging in semantic retrieval to show the structure of the semantic memory compression.*
>
> We appreciate the suggestion regarding visualizations of semantics and episodes.
>
> We provide visualizations of episodes and semantics extracted by ECO and SeTR, respectively, from a sample video (https://gifyu.com/image/SGhZw)
> * Episode capturing the flies: https://gifyu.com/image/SGhYV
> * Episode capturing the landscape: https://gifyu.com/image/SGhY6
> * Episode capturing flying birds: https://gifyu.com/image/SGhY4
> * Semantic representation of the land: https://gifyu.com/image/SGhrs
> * Semantic representation of the lizard: https://gifyu.com/image/SGhrL
> * Semantic representation of the birds: https://gifyu.com/image/SGhrp
>
> We will add these visualizations in the revised version of our paper to help the reader better understand how HERMES maintains coherent episodic and semantic representations while reducing dimensionality.
>
> ### Reply to: *The results on more recent long video benchmarks, e.g., VideoMME, MLVU, LVBench, etc, are desired.*
>
> We appreciate the reviewers' suggestions regarding newer benchmarks. We want to clarify important context about our work's positioning and contributions:
>
> 1. Focus on Resource Efficiency:
> - Our work mainly targets training-free improvements through ECO and SeTR modules
> - These modules provide significant gains without requiring expensive pretraining or massive compute resources (which we don't have)
> - This approach makes our contribution accessible to broader research community as they can be used as inexpensive plug-ins
> - Table 7 demonstrates this: substantial improvements to MA-LMM through plug-and-play integration
>
> 2. Benchmark Selection Rationale:
> - Our current backbone (BLIP-2) was pretrained on images only and on much smaller datasets than the most recent video-language models. This makes HERMES more suitable for traditional video benchmarks (LVU, Breakfast, COIN), especially for fully-supervised tasks
> - We have strong results (zero-shot and fully-supervised) on MovieChat-1k which shows our modules' effectiveness even with a simpler backbone. Datasets such as VideoMME, MLVU and LVBench are mostly MCQs and zero-shot only, therefore our backbone is not suitable for them. We recognize that this is a weakness of our model, and we are working on addressing it.
>
> 3. How do we address it?
> - We designed ECO and SeTR to be model-agnostic (we showed a glimpse of that in Table 7)
> - For camera-ready, we will demonstrate portability to newer models and test them on VideoMME, MLVU and LVBench. Such integration will address the backbone weakness we talked about earlier and showcase our modules' value as universal improvements to video-language models.
>
> We look forward to your feedback and are happy to provide any additional information or analyses that would be helpful in your evaluation. Thank you.

---

> > ### Author Response · Authors · 2024-11-25
> >
> > We deeply appreciate your thoughtful and constructive review, which has been instrumental in refining HERMES's conceptual and technical foundations.
> >
> > Your feedback on our architectural design has prompted us to:
> >
> > - Provide more nuanced visualizations of episodic and semantic memory extraction
> > - Clarify our cognitive-inspired approach to video understanding
> > - Demonstrate the model-agnostic nature of our ECO and SeTR modules
> > - Better position our work as an efficient way to gain training-free improvement (making it more accessible to researchers with limited computational budget)
> >
> >
> > Commitment:
> > - Demonstrate portability across different video-language models to address the limitations of our current image-pretrained backbone
> >
> > We respectfully request:
> >
> > - A reconsideration of the initial score in light of these substantive improvements
> > - An opportunity to provide any additional clarifications during the remaining discussion period

---

### Official Review · Reviewer_BVft · 2024-11-04

**Soundness:** 2
**Presentation:** 3
**Contribution:** 2
**Rating:** 3
**Confidence:** 4

**Summary:**

The paper presents HERMES, a framework designed for long-form video understanding that leverages episodic memory and semantic knowledge. The authors propose two main components: the Episodic COmpressor (ECO), which aggregates temporal information, and the Semantics reTRiever (SeTR), which extracts high-level semantic cues. The model claims to outperform existing methods on multiple benchmarks, showcasing its effectiveness in handling long videos.

**Strengths:**

- The authors enhance Q-former-based models, such as MA-LMM, by integrating episodic and semantic information. The architecture is clearly defined and appears to be replicable.
- The paper reports impressive performance metrics across various long-video benchmarks, indicating that HERMES effectively addresses the complexities inherent in long-form video content.

**Weaknesses:**

- While the use of memory mechanisms and temporal information compression is valuable, the approach does not significantly advance the current state of the art. The paper primarily builds on the MA-LMM framework, with its main innovations centered around the memory components. The concepts introduced in the Episodic COmpressor and Semantics reTRiever share similarities with the memory bank and mechanisms used in MA-LMM. Furthermore, comparable memory structures are present in MovieChat, which differentiates between long-term and short-term memory—concepts that correspond to the SeTR and ECO, respectively. The authors are encouraged to discuss these distinctions in detail to better highlight the unique contributions of HERMES.
- The methods used for comparison across different datasets are inconsistent, making it difficult to draw clear conclusions about HERMES's overall performance. A standardized evaluation framework would enhance the reliability of the results.
- There are inconsistencies in the use of symbols throughout the paper. For instance, in Line 208, \( F_w \) refers to the number of frame features, while in Line 222, \( F_w \) denotes the window, with \( w \) indicating the number of frame features. This inconsistency could lead to confusion and should be clarified.

**Questions:**

Can the authors provide a more detailed discussion on how HERMES distinguishes itself from existing models like MA-LMM and MovieChat? Specifically, what unique contributions do the Episodic COmpressor (ECO) and Semantics reTRiever (SeTR) offer compared to similar mechanisms in these models?

---

> ### Author Response · Authors · 2024-11-23
>
> Thank you for your thoughtful review and insightful questions. We appreciate the opportunity to clarify HERMES's unique contributions and address your concerns.
>
>
> ## Memory Management Differences between HERMES, MA-LMM and MovieChat
> - HERMES versus MA-LMM: For each incoming frame, MA-LMM adds it to the memory bank by computing the similarities with adjacent frames and merging the incoming frame with its most similar in the memory bank. Below are our main differences.
>   * **HERMES takes a distributed approach**. Our ECO, distributes the frames of the incoming window to the most appropriate episode. This approach is more intuitive and better mirrors human memory formation:
>     - Frames can be grouped into episodes regardless of temporal adjacency
>     - This naturally handles scene transitions, flashbacks, and non-linear narratives
>   * **HERMES is vastly more efficient and accurate**. As shown in Table 7 in the paper, our memory management system almost halves the inference time (-43%) when plugged into MA-LMM while being 3.4% more accurate.
>   * **HERMES also captures semantics**. Our Semantics Retriever (SeTR) complements the episodic memory and is shown (in Table 7) to increase the accuracy of MA-LMM by almost 4% with only a negligible increase in latency.
>
> - HERMES versus MovieChat: Moviechat's short-term memory uses a FIFO mechanism. Its long-term memory uses ToMe. Below are the main differences
>     * **HERMES has episodes** instead of a short term memory, and our update approach is based on similarity to a certain existing episode instead of FIFO. As shown in Table 4 of the paper, FIFO's performance is inferior to ECO.
>     * **HERMES's long-term memory is implicitly encoded** in ECO. We consider SeTR as a semantics scanner that retrieves scattered semantics from the video.
>     * **22 FPS processing speed** compared to MovieChat's ~0.01 FPS (1 day vs. 13 minutes on MovieChat-1k) using a V100 GPU (32 GB).
>     * Uses only 100 frames vs MovieChat's 2048 frames (detailed analysis as to why in Appendix A.4.1)
>
> ## Evaluation Methodology Clarification
> We follow standard evaluation protocols for each dataset:
> - LVU, Breakfast, COIN: Top-1 accuracy
> - MovieChat-1k: GPT-assisted scores
>
> ## Symbol Clarification
> We thank you for catching the \(F_w\) inconsistency. We have revised the notation:
> - Line 208: "Upon receiving a window of frame features, \(F_w\), ..."
> - Line 222: "\(F_w\) represents the incoming window of frame features..."
>
> ## (Extra) Visualizing Episodes and Semantics
> We provide visualizations of episodes and semantics extracted by ECO and SeTR respectively from a sample video (https://gifyu.com/image/SGhZw)
> * Episode capturing the flies: https://gifyu.com/image/SGhYV
> * Episode capturing the landscape: https://gifyu.com/image/SGhY6
> * Episode capturing flying birds: https://gifyu.com/image/SGhY4
> * Semantic representation of the land: https://gifyu.com/image/SGhrs
> * Semantic representation of the lizard: https://gifyu.com/image/SGhrL
> * Semantic representation of the birds: https://gifyu.com/image/SGhrp
>
>
> Given HERMES's significant efficiency gains and novel architectural approach, we believe it represents more than an incremental advancement. We look forward to your feedback and are happy to provide any additional information or analyses that would be helpful in your evaluation. Thank you.

---

> ### Author Response · Authors · 2024-11-25
>
> In light of our response addressing your specific concerns and demonstrated commitment to improving the work, we respectfully request a reconsideration of the initial score, given the following improvements:
>
> - Demonstrated performance gains and efficiency improvements
> - Demonstrated Distinctions from Existing Works
> - Our clarifications about the methodology
> - Changes to the manuscript addressing the notation inconsistency
>
> If any remaining uncertainties persist, we are prepared to:
>
> - Provide additional clarifications
> - Conduct supplementary experiments
> - Discuss any nuanced technical aspects you find intriguing or require further elaboration
>
> We appreciate your time, thoroughness, and the opportunity to improve our work through this discussion.

---

### Note · Authors · 2024-11-30

**Comment:**

We are grateful for the reviewers' constructive feedback on our paper and ideas for improvement.

**Withdrawal Confirmation:**

I have read and agree with the venue's withdrawal policy on behalf of myself and my co-authors.